# Gender-Based Violence Perpetration by Male High School Students in Eastern Ethiopia

**DOI:** 10.3390/ijerph17155536

**Published:** 2020-07-31

**Authors:** Addisu Shunu Beyene, Catherine Chojenta, Deborah J. Loxton

**Affiliations:** 1Research Centre for Generational Health and Ageing, Faculty of Health and Medicine, University of Newcastle, Newcastle, NSW 2308, Australia; catherine.chojenta@newcastle.edu.au (C.C.); deborah.loxton@newcastle.edu.au (D.J.L.); 2School of Public Health, College of Health and Medical Sciences, Haramaya University, Harar 235, Ethiopia

**Keywords:** gender-based violence, perpetration, male, high school, students, Ethiopia

## Abstract

Gender-based violence (GBV) perpetration is a global public health problem due to its detrimental effect on health and education. This study aims to determine the prevalence of gender-based violence perpetration by male students in eastern Ethiopia. A cross-sectional study was conducted in eastern Ethiopia in December 2018. A total of 1064 male students were involved in the study. Data were collected using an adaptation of the WHO Multi-Country Study self-administered questionnaire on the Women Health and Life Event. Descriptive statistics were calculated using STATA version 14. The prevalence of gender-based violence committed by a male in the last 12 months was 55.83% (95% CI: 52.84–58.82%). The prevalence of emotional abuse against an intimate or non-partner was 45.86% (95% CI: 42.87–48.86%), physical abuse was 45.77% (95% CI: 42.77–48.77%), and sexual abuse was 31.11% (95% CI: 28.32–33.90%). The perpetration of multiple types of gender-based violence (emotional, physical, and sexual) was 47.15% (95% CI: 43.15–51.25%), with 17.72% (95% CI: 14.75–21.03%) reporting emotionally and physically violent acts, 14.21% (95% CI: 11.51–17.27%) reporting emotionally violent acts only, and 12.88% (95% CI: 10.29–15.82%) reporting physically violent acts only. There were statistically significant differences between the age of participants who committed acts of all forms of GBV in the “ever” timeframe and the past 12 months (*p* < 0.001). Effective prevention and intervention strategies should be developed at the school level to reduce gender-based violence perpetration.

## 1. Introduction

Gender-based violence (GBV) perpetration is a global public health problem due to its detrimental effect on health and education, including depression, injuries, self-harm, sexually transmitted infections, substance misuse, chronic diseases [1,2,3], low school achievement, an increased school dropout rate, and increased absenteeism [4,5]. GBV is linked to gender inequalities, the absence of power and control, social norms, and the condoning of abuse. It creates and maintains the imbalance of power between men and women and is a fundamental violation of human rights [6,7].

Despite its prevalence globally, available studies on GBV perpetration by males have mainly been conducted in developed countries and a limited number of other settings [8,9,10,11]. The available studies on GBV perpetration by males are limited, especially in low- and middle-income countries, and GBV has been minimally researched in school settings [12]. In a study conducted by the World Health Organization, it was found that schools and universities were highly vulnerable settings for GBV [12]. However, this problem has not been well investigated in educational institutions and GBV in a school setting has been noted as a significant public health problem [13,14].

Importantly, we found few studies in Africa that focused on GBV perpetration by males [11,15,16,17,18,19,20,21]. These studies of male participants revealed that the prevalence of overall GBV was 24.4% to 51.5% [11,18,19,21]; for emotional violence it was 23.1% to 42.2% [17,18,19], for physical violence it was 11.9% to 46.2% [11,17,18,19,21], and for sexual violence it was 11.7% to 70% [11,15,16,17,19,20,21]. However, the findings of these studies varied greatly due to the sampling, definitions used, settings, and cultures.

Like other African countries, Ethiopia is a patriarchal society with rigid gender norms and cultural values. In this type of society, GBV is mostly supported by cultural norms and institutions, which support male dominance and sexual entitlement and expectations for females to be submissive [10,16,22,23]. Ethiopia is also one of the many African countries which has a lack of data on GBV perpetration. The majority of studies conducted in Ethiopia have been focused on reports by females [24,25,26,27,28], except for two studies of GBV acts by males [11,16]. These two studies noted that the prevalence of GBV acts by male students was between 24.4% and 70.5% [11,16]. However, these two studies did not measure all types of GBV perpetration. Bekele et al. exclusively focused on sexual violence committed by male school students. However, this study did not measure other types of GBV and current sexual violence [16]. Philpart et al. measured sexual and physical violence committed by male college students. However, this study did not measure emotional violence and lifetime prevalence [11]. Thus, our study was focused on all types of GBV perpetration and its overall occurrence (emotional, physical, and sexual violence) during the perpetrator’s lifetime and during the previous 12 months among high school students. Moreover, most of the above studies have only reported one or two different forms of GBV. This highlights the importance of documenting evidence on how males report their violent behavior against females. To date, no study has been examined all types of GBV and its overall prevalence among male high school students in Ethiopia. Filling these gaps and having more information is needed in order to design appropriate interventions strategies to reduce GBV in schools.

Therefore, determining the prevalence of the different forms of GBV perpetration by males attending school in Ethiopia will show the scope of the issue to be addressed and help policymakers to develop interventions. This study aimed to determine the magnitude of the different types of GBV perpetration by male students in eastern Ethiopia.

## 2. Materials and Methods

### 2.1. Study Area, Design, and Sampling

An institutional-based cross-sectional survey was conducted in December 2018 in the East Hararghe Zone, which is 525 kilometers east of Ethiopia’s capital, Addis Ababa. A single population proportion formula was used to calculate the sample size using Epi Info version 7, taking into consideration the scientific assumptions and all the objectives of the study. The objectives which provided the maximum sample were taken. The study used a multi-stage sampling technique. First, the total number of schools and students was obtained from the zone education office. In the second stage, five schools were selected using simple random sampling from the total number of schools in the zone. In the third stage, classes from each grade level were selected using simple random sampling, and all the male students from those classes were invited to participate. The male students who were absent during data collection were excluded. The study sample was proportionally allocated to each selected school based on their size. A total of 1109 male students aged 15–24 years were invited. Girls were included in this procedure, but their data are not included in this paper.

### 2.2. Data Collection Methods and Tools

Data were collected using pretested self-administered questionnaires adapted from the World Health Organization’s (WHO) Multi-Country Study on Women’s Health and Life Events, the revised Conflict Tact Scales (CTS2), and the Ethiopian Demographic and Health Survey (EDHS) [22,29,30], and adapted in previous studies for use in Ethiopia [11,22,31]. The questionnaire was adapted from previous studies for use in Ethiopia to ask about the perpetration of violence by males (the original questions were about experience of GBV) [31]. The questionnaire was translated into Afaan Oromo (the local language) and back into English by independent translators to ensure consistent translation. The questionnaires were revised based on the translations and skipping patterns. The data collectors were qualified male health professionals (e.g., nursing, public health, psychiatry) and all could speak the local language. The survey was undertaken in a separate classroom with the help of data collectors and supervisors. The survey took 40–60 min to finish.

### 2.3. Study Variables

The outcomes variables were all types of GBV perpetration (physical, sexual, and emotional violent acts). We measured the different types of GBV using the adapted WHO-Multi Country Study Survey and EDHS and adapted it in a previous study for use in Ethiopia. For each item, the participants were asked if they had once, a few times, or many times committed a GBV act in the “ever” timeframe and in the past 12 months. We measured any type of GBV and its items as follows:

Physically violent behaviors were assessed using six items: (1) throwing things/slapping; (2) shoving/pushing (which were categorized as moderate physically abusive behaviors); (3) burning; (4) beating; (5) kicking/dragging; and (6) intimidating using a weapon (e.g., gun, knife, or another object). The last four items were categorized as severe physically abusive behaviors. At least one “yes” answer to the six items qualified the person as committing a physically violent act [11,22,29,31].

Sexually abusive behaviors were measured using three items: (1) physically forcing a woman to have sexual intercourse; (2) forcing a woman to perform any other sexual acts; and (3) forcing a woman with threats or in any other way to perform sexual acts without her consent (degraded/humiliated). At least one “yes” answer to the three items qualified the person as committing a sexually violent act [11,22,29,31].

Emotional violence was assessed using four items: (1) insulted/made a woman feel bad about herself; (2) belittled or humiliated a woman in front of others; (3) scared/intimidated a woman on purpose (e.g., by the way they looked at her, by yelling, or with threats); and (4) threatened a woman/someone she cares about. At least one “yes” answer to the four items qualified the person as committing an emotionally violent act [11,22,29,31].

All the items were answered using no (0) or yes (1). The perpetration of physical, sexual, and emotional violence was measured for the previous 12 months and in the “ever” timeframe. The prevalence of each item was calculated separately and restricted to the “ever” timeframe and the past 12 months. The prevalence of any GBV was measured as the perpetration of at least one or more acts of emotionally, physically, or sexually violent acts committed by a male against a partner or non-partner. To estimate the prevalence of GBV, emotional violence, physical violence, and sexual violence were coded into dichotomous variables (1: yes; 0: no). These questionnaires were adapted in previous studies for use in Ethiopia [11,22,29,31].

Information regarding socio-demographic variables, including age, educational status (Grade 9, Grade 10, Grade 11, or Grade 12), childhood residence (urban or rural), ethnicity (Oromo, Amhara, Gurage, Somali, or other), religion (Orthodox, Protestant, Muslim, Catholic, or other), current living situation (living with family, male/female friend, relatives, or others), marital status (married, girlfriend, or neither) was also collected. We measured academic performance based on the student’s last semester cumulative grade points and the student’s self-rated academic performance as good, very good, or poor [11,16,22].

### 2.4. Data Quality Control

Three days of intensive training were provided for the data collectors and supervisors with an emphasis on the study objective, sampling methods, tools, data collection methods, ethical issues of GBV research, details in the information statement, sensitive questions, and the data quality assurance. A pre-test was conducted on 5% of the sample size in one of the non-selected schools in East Hararghe. The questionnaires were revised to correct the skipping patterns. Supportive supervision was undertaken during field work by the principal investigator. The study participants were well informed about the aims and importance of the study, thereby creating a conducive environment to minimize their concerns, as the study touches on sensitive issues. We executed descriptive statistics including frequencies, sorting, and cross tabulations to check the outliers, inconsistencies, and missing values before the data analysis. We also presented the data using tables to check for outliers, logical inconsistencies, and patterns of missing information. Cross checking of the hard copy and electronic versions of the survey data was also carried out. After we checked the data, the variables with missing data due to missing skipping patterns were managed by recoding the missing value to the existing variable. The categories of variables with small cell counts were collapsed into a single category.

### 2.5. Ethical Considerations

Ethical clearance was secured from the University of Newcastle Human Research Ethics Committee (H-2018-0031) and Haramaya University, College of Health and Medicine Science (Ref. No. IHRERC/137/2018). Official permission in writing was obtained from the Zonal Education Office Representatives, the district office, and the selected schools’ principals. As the study involved a single survey and did not involve the collection of personally identifiable information, the ethics committee recommended that formal consent from the participant was not required. Consent was implied by the completion of the survey. This process would enhance the assurances of anonymity given to the students. Students could decide not to take part by not attending the survey session, or not completing a survey within the survey session (by placing a blank or incomplete survey in the provided envelope). Reasons for non-participation or withdrawal were not required, and there were no consequences for participation or non-participation. The study strictly followed the WHO guidelines on ethical issues related to the study of gender-based violence [32,33]. All the information obtained from the respondents remained confidential and anonymous.

### 2.6. Data Analyses

The collected data were cleaned and entered to Epidata version 3.1.0 and exported to STATA 14 for analysis. All the analyses were conducted using STATA version 14 and 16. The collected data were checked for outliers and missing values. We assume that all the school students shared similar characteristics (age, grade). We also proportionally allocated the sample size to the selected schools based on their size. Because of this, we did not weigh the data. Descriptive statistics were calculated for the student’s socio-demographic variables. The prevalence of perpetration of any type of GBV and different forms of GBV were determined separately using a 95% CI. A chi-square test was used to assess the differences between age with any type of GBV.

## 3. Results

### 3.1. Socio-Demographic Characteristics

A total of 1064 participants took part in the study, with a response rate of 96.5%. The majority of the respondents (89.94%) were Muslim, and 69.92% were in the age range of 16–19 years, with a mean age of 17.28 years. Almost all the respondents (96.52%) were of Oromo ethnicity, and 54.70% resided in a rural area before they were 12 years old. More than half (67.89%) of the respondents were currently living with their parents. Among the study participants, 44.08% were attending Grade 9, 54.32% had an average academic performance, and 70.77% of the respondents had a girlfriend at the time of the survey (Table 1).

### 3.2. Prevalence of Any Gender-Based Violence (GBV) and Overlap

The prevalence of perpetrating any type of GBV in the “ever” timeframe was 58.93% (95% CI: 55.97–61.89) and over the last 12 months, 55.83% (95% CI: 52.84–58.82; Table 2).

A substantial overlap of perpetrating multiple types of GBV was reported by the participants. In the “ever” timeframe, out of those who reported perpetrating any type of GBV, 46.13% (95% CI: 42.19–50.10) reported perpetrating all types of GBV acts (emotional, physical, and sexual), 19.12% (95% CI: 16.12–22.39) reported both emotionally and physically violent acts, 2.84% (95% CI: 1.69–4.46) reported emotionally and sexually violent acts, 2.84% (95% CI: 1.69–4.46) reported both physically and sexually violent acts, 13.59% (95% CI: 11.01–16.50) reported emotionally violent acts only, 13.90% (95% CI: 11.30–16.84) reported physically violent acts only, and 1.58% (95% CI: 0.76–2.89) reported sexually violent acts only (Figure 1).

Over the previous 12 months, among those who reported perpetrating any type of GBV, 47.15% (95% CI: 43.15–51.25) reported all GBV violent acts (emotional, physical and sexual), 17.72% (95% CI: 14.75–21.03) reported emotionally and physically violent acts, 2.51% (95% CI: 1.41–4.10) reported emotionally and sexually violent acts, 3.51% (95% CI: 2.19–5.32) reported physically and sexually violent acts, 14.21% (95% CI: 11.51–17.27) reported emotionally violent acts only, 12.88% (95% CI: 10.29–15.82) reported physically violent acts only, and 2.01% (95% CI: 1.04–3.48) reported sexually violent acts only.

### 3.3. Prevalence of Emotional Violence

One out of three (35.71%) males reported that in the “ever” timeframe they had insulted a woman or made a woman feel bad. About 38% of males had ever humiliated a woman in front of other people. Additionally, 34.30% had intimidated a woman on purpose, and 34.02% had threatened someone a woman cared about. Overall, the prevalence of emotionally abusing women in the “ever” timeframe was 48.59% (95% CI: 45.58–51.60), and over the past 12 months it was 45.86% (95% CI: 42.87–48.86; Table 2).

### 3.4. Prevalence of Physical Violence

About 32% of males had ever slapped/thrown something at a woman, and 37.69% had ever pushed/shoved a woman. These are common acts of moderate physical violence. Of the males who reported having committed physical violence, 26.88% (95% CI: 24.21–29.55) and 25.47% (95% CI: 22.85–28.09) committed moderate physical abuse against women in the “ever” timeframe and over the past 12 months, respectively. Common acts of severe physical violence were kicking/dragging, beating with a fist, choking/burning, and threatening with a weapon. One fifth (20.21%; 95% CI: 17.79–22.62%) of respondents committed severe physical violence against women in the “ever” timeframe. Generally, the prevalence of perpetrating physical abuse in the “ever” timeframe was 48.87% (95% CI: 45.86–51.88), and over the past 12 months it was 45.77% (95% CI: 42.77–48.77; Table 2).

### 3.5. Prevalence of Sexual Violence

Almost a quarter (23.03%) of participants reported committing acts of forced sexual intercourse against women in the “ever” timeframe. Meanwhile, 29.51% of males committed other forced sexual acts against women in the “ever” timeframe. In the “ever” timeframe, nearly a quarter (24.25%) of respondents committed forced sexual acts against women that degraded/humiliated the women. Overall, the prevalence of sexual violence committed by males against women in the “ever” timeframe was 31.86% (95% CI: 29.07–34.67), and over the last 12 months it was 31.11% (95% CI: 28.32–33.90; Table 2).

### 3.6. Differences in All Forms of Gender-Based Violence (GBV)

Differences in demographic features were observed in all forms of GBV in the “ever” timeframe and in the past 12 months. There were statistically significant differences between the age of participants who committed acts of all forms of GBV in the “ever” timeframe and in the past 12 months (*p* < 0.001). There were also statistically significant differences between the educational levels of participants who committed acts of sexual violence over the past 12 months (*p* < 0.001). However, there were no statistically significant differences between the educational levels of participants who committed acts of physical, emotional, or any GBV in the “ever” timeframe and in the past 12 months. Likewise, there were no statistically significant differences between the educational levels of participants who committed sexual violence in the “ever” timeframe (Table 3).

## 4. Discussion

This is one of the first school-based studies to investigate the “ever” and 12-month prevalence of different forms of GBV perpetration in Ethiopia. It adds important information to available literature concerning the prevalence of GBV and the overlapping of emotional, physical, and sexual violence in the “ever” timeframe and in the past 12 months. In this study, the prevalence of any type of GBV in a male student’s “ever” timeframe was 58.93%, and over the past 12 months it was 55.83%. This finding is higher than the studies conducted in Hawasa, Ethiopia (24.40%; [11]); South Africa (31.1%; [19]); and Ghana (40.5%; [18]). This high prevalence may be due to male students being surveyed by male data collectors who were familiar with the culture of the study area. Emotional/psychological violence was included in this study and had not been previously studied in Ethiopia. This may account for the higher prevalence of GBV in this study compared to previous studies. Another possible explanation may be the time reference. For example, the current study measured GBV in the “ever” timeframe and over the past 12 months, whereas other studies measured differing time periods. Additionally, the sampled populations were varied (high school, college, university, urban, rural), as were the definitions of GBV that were used (single or two items). Other differences could be attributed to age differences between the study populations.

The high prevalence of GBV by male students could be explained by the modern and traditional culture in Ethiopian society that values men over women. Males are expected to have sexual knowledge and be sexually active, whereas for females this would be considered taboo because of religious, social, and cultural norms in Ethiopian society. This reflects gender inequality, which is strongly associated with GBV [14,34,35]. Moreover, societies with an ideology of male superiority have higher rates of GBV [34,36].

In the current study, the prevalence of perpetrating emotional violence in a participant’s “ever” timeframe was 48.59%, and over the last 12 months it was 45.86%. This result is consistent with findings from South Africa (42.2%; [19]), but higher than findings from Ghana (23.1%; [18]) and another study in Cape Town, South Africa (31.2%; [17]). The differences in prevalence might be due to the different definitions used, target populations, study settings, and socio-cultural norms, and highlight the need for country- and region-specific studies. Other reasons for the higher prevalence of emotional abuse may be that perpetration of violence against women in Ethiopia is deeply entrenched and accepted [31,34,35].

This study showed that the prevalence of perpetrating physical violence in a participant’s “ever” timeframe was 48.87%, and over the past 12 months it was 45.77%. For the period of the past 12 months, 20.21% reported severe physically violent acts and 25.47% reported moderate physically violent acts. The prevalence of physical abuse corroborates a study conducted in South Africa (42.3%) [19] but is higher than previous results from Ethiopia (15.8%; [11]); Cape Town, South Africa (20.3%; [17]); and Ghana (11.9%; [18]). This difference might be due to the sampled population, time difference, definitions used, or age differences between the study populations. For instance, the present study assessed physical violence during the participant’s lifetime and over the past 12 months against partners or non-partners, whereas the previous studies used different timeframe [17,18].

The present study indicated that the prevalence of sexually violent acts in a participant’s “ever” timeframe was 31.86%, and over the past 12 months it was 31.11%. This result is lower than a finding in eastern Ethiopia (70.4%; [16]), but higher than the other findings in southern Ethiopia (16.9%; [11]); South Africa (15.3%; [15,19]); Cape Town, South Africa (11.0%; [17]); Spain (15%; [20]); and Ghana (16.6%; [18]). The discrepancy might be due to varying definitions used, setting, age differences, and the sampled population. For example, the current study used three items to measure sexual violence, but the previous studies used a single or two items to measure sexual violence. Additionally, the samples were drawn from different populations.

In general, the present study demonstrated that GBV perpetration is highly prevalent in Ethiopian school settings. This high prevalence could be explained by the presence of a traditional culture in the study area that supports the superiority of men and the acceptability of males using violence against their partners in Ethiopia [16,22,23,34]. The perpetration of violence against women in Ethiopia has been found to be deeply entrenched and accepted [31,34,35]. Male sexual entitlement and males’ belief in their right to decide when and how to have sex also play a role [11,16,23,28,34,35,37,38].

Moreover, the findings of this study have paramount importance for the Ethiopian community and Sub-Saharan Africa. It creates an awareness about the prevalence of GBV perpetration by male students. This information can be the groundwork for the development of appropriate intervention to minimize the burdens of gender-based violence at the school level [39]. It may help to develop or modify interventions that other youths in the school might benefit from in the future [39]. It can also serve as a baseline for conducting further study in this area.

To the best of the authors’ knowledge, this is the first study to investigate the sizable co-occurrence of the three forms of GBV in an Ethiopian school setting. The most common joint occurrence was emotionally, physically, and sexually violent acts, followed by physically and emotionally violent acts. The high prevalence of co-occurring forms of GBV suggests that male students in schools are more likely to perpetrate multiple forms of violent acts than emotionally violent acts. This finding suggests that primary prevention and appropriate intervention strategies should consider all forms of violence and be developed for application at the school level [39,40,41].

Males over 15 years reported significantly higher proportions of all forms of GBV than those aged under 15 years. This finding indicates that as males get older, they are more likely to perpetrate GBV. This result reveals that prevention strategies might be best aimed at males under 15 years, with both prevention and intervention policies designed for males aged over 15 years.

## 5. Limitations of This Study

This study was not free of limitations. We were unable to include out of school youths; respondents may have either exaggerated or underreported their perpetration due to recall bias, social desirability bias, guilt through undertaking actions that they might perceive as wrong, or non-compliance; and our findings cannot be generalized to all males in Ethiopia. However, the study does provide vital evidence and some guidance for the prevention of GBV perpetration by male students in eastern Ethiopia.

## 6. Public Health Implications

This study revealed that GBV perpetration is a common problem in high schools. This suggests that GBV in schools needs urgent attention and intervention from government officials, NGOs, and other organizations. Therefore, to minimize the high prevalence of GBV perpetration in schools, the researchers have the following suggestions: (1) Incorporating the magnitude of the GBV, its types, and its consequences into school curricula might assist in the minimization of GBV. Creating youth awareness about the prevalence, risk factors, and consequences of GBV may assist in reducing the occurrence of GBV [39,40,42]. (2) Schools should develop a zero-tolerance policy that prevents students from committing any form of GBV when at school or involved in school activities [39,40,41]. (3) Schools could encourage students to organize an event to raise the awareness of GBV (gender club, mass media, culture and norms, training, and education). (4) Educate the community and parents about the prevalence, causes, and consequences of GBV through information, education, or behavior change communication (IE/BCC) [39,40,41,42]. (5) Given the extensive scope of GBV seen in current research, schools should create collaboration between different stakeholders, including teachers, parents, the education office, the justice system, the health office, the legal system, the police, and NGOs to prevent GBV. (6) Education office/schools should encourage and expand the culture and social norms that support non-violence and gender equitable relationships and promote women’s empowerment at the school and community level [39,40,41].

Future research directions: (i) national-level longitudinal studies are needed to thoroughly investigate the cross-cultural factors, consequences, and possible prevention mechanisms for GBV (in school and out of school); (ii) examining GBV and its associated factors among out-of-school male youths in Ethiopia; and (iii) investigating risk and protective factors for the co-occurrence of multiple forms of GBV among youths (in school and out of school).

## 7. Conclusions

This study showed that the prevalence of all forms of GBV perpetration by male students in Ethiopia was high. A substantial number of students reported the co-occurrence of different forms of GBV. The results of this study have implications for school-based policies and interventions designed to prevent GBV acts by males in Ethiopia [40,41]. This reveals that prevention should be the focus for males under 15 years and both prevention and intervention activities for males over 15 years. Additionally, the prevention of GBV should be focused on changing cultural and social norms around male dominance and sexual entitlement [40,41].

## Figures and Tables

**Figure 1 ijerph-17-05536-f001:**
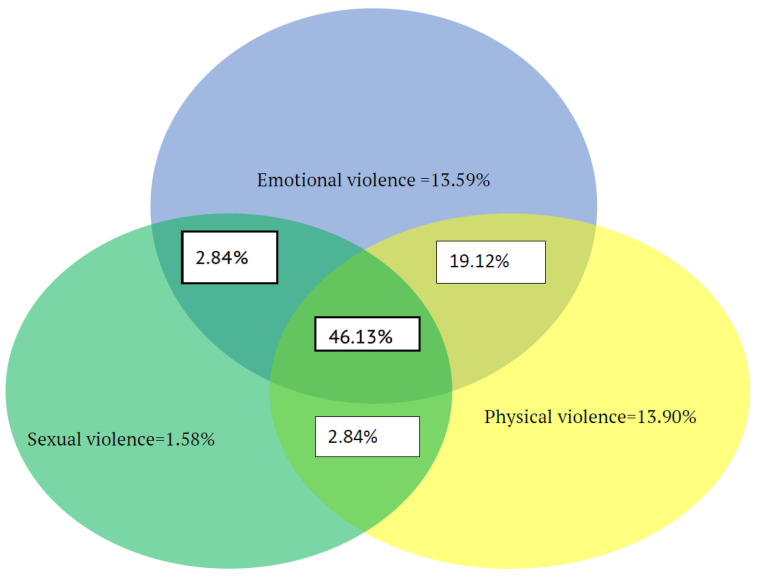
Venn diagram showing the overlap of types of gender-based violence (GBV) committed by male high school students during their lifetime in East Hararghe, eastern Ethiopia, December 2018. The size of the circles is not proportional to the percentage of each type of GBV.

**Table 1 ijerph-17-05536-t001:** Socio-demographic characteristics of the male respondents in East Hararghe, eastern Ethiopia, December 2018.

Variables	Frequency (N)	Percentage (%)
Age		
≤15	196	18.42
16–19	744	69.92
≥20	124	11.65
Religion		
Orthodox	81	7.61
Catholic	5	0.47
Protestant	20	1.88
Muslim	957	89.94
Other	1	0.09
Ethnicity		
Oromo	1027	96.52
Amhara	20	1.88
Gurage	9	0.85
Somali	3	0.28
Other ^b^	5	0.47
Childhood residency		
Urban	482	45.30
Rural	582	54.70
Current living arrangements ^a^		
Alone	103	9.70
With family	721	67.89
With male friends	287	27.02
With female friends	285	26.84
With relatives	87	8.19
With others	3	0.28
Education		
Grade 9	469	44.08
Grade 10	346	32.52
Grade 11	161	15.13
Grade 12	88	8.27
Academic performance		
Good and above	470	44.17
Average	578	54.32
Poor	16	1.50
Ever married		
Yes	89	8.36
No	975	91.64
Partner status		
Yes, married	37	3.48
Yes, girlfriend	753	70.77
No	274	25.75
Partner’s education (*n* = 790)		
Illiterate	21	2.66
Grades 1–8 complete	222	28.10
Grades 9–12 complete	348	44.05
Above Grade 12 complete	129	16.33
I don’t know	70	8.86
Partner’s occupation (*n* = 790)		
Student	717	90.53
Government employee	27	3.42
Unemployed	22	2.78
Merchant	21	2.65
Other ^c^	5	0.53

^a^ Multiple response; ^b^ Harari, Silte; ^c^ daily laborer, driver, self-employed, and janitor. Partner for education occupation questions: those who were married or had girlfriends.

**Table 2 ijerph-17-05536-t002:** Type and prevalence of GBV perpetrated by male high school students in the “ever” timeframe and over the previous 12 months, eastern Hararghe, eastern Ethiopia, December 2018.

Types of Gender-based violence	“Ever”	Frequency in the “Ever” Timeframe ^a^			Past 12 Months	Frequency in the Last 12 Months ^b^		
	Number %	Once	A Few Times	Many Times	Number %	Once	A Few Times	Many Times,
Emotional/psychological violence								
Insulted/made to feel bad	380 (35.71)	182 (47.89)	119 (31.32)	79 (20.79)	352 (33.08)	174 (49.43)	99 (28.13)	79 (22.44)
Humiliated in front of others	405 (38.06)	195 (48.15)	126 (31.11)	84 (20.74)	376 (35.34)	189 (50.27)	117 (31.12)	70 (18.62)
Intimidated on purpose	365 (34.30)	189 (51.78)	102 (27.95)	74 (20.27)	345 (32.42)	174 (50.43)	103 (29.86)	68 (19.71)
Threatened/hurt someone a woman cares about	362 (34.02)	190 (52.49)	109 (30.11)	63 (17.40)	344 (32.33)	189 (54.94)	103 (29.94)	52 (15.12)
At least one incident of emotional violence	**517 (48.59)**	**292 (56.48)**	**153 (29.59)**	**72 (13.93)**	**488 (45.86)**	**267 (54.71)**	**147 (30.12)**	**74 (15.16)**
Physical violence								
Moderate physical violence	286 (26.88)	193 (42.70)	175 (38.72)	84 (18.58)	271 (25.47)	190 (45.02)	153 (36.36)	79 (18.72)
Slapped/thrown something	337 (31.67)	161 (47.77)	119 (35.31)	57 (16.91)	319 (29.98)	162 (50.78)	99 (31.03)	58 (18.18)
Pushed/shoved	401 (37.69)	198 (49.38)	147 (36.66)	56 (13.97)	374 (35.15)	188 (50.27)	137 (36.63)	49 (13.10)
Severe physical violence	215 (20.21)	156 (36.79)	162 (38.21)	106 (25.00)	215 (20.21)	152 (37.53)	146 (36.05)	107 (26.42)
Beat with a fist/something else	306 (28.76)	153 (50.00)	102 (33.33)	51 (16.67)	303 (28.48)	144 (47.52)	106 (34.98)	53 (17.49)
Kicked/dragged	368 (34.59)	158 (42.93)	141 (38.32)	69 (18.75)	350 (32.89)	161 (46.00)	128 (36.57)	61 (17.43)
Choked/burnt	296 (27.82)	160 (54.05)	88 (29.73)	48 (16.22)	283 (26.60)	156 (55.12)	85 (30.04)	42 (14.84)
Threatened or used a weapon (e.g., gun or knife)	259 (24.34)	133 (51.35)	77 (29.73)	49 (18.92)	261 (24.53)	131 (50.19)	76 (29.12)	54 (20.69)
At least one incident of physical violence	**520 (48.87)**	**299 (57.50)**	**155 (29.81)**	**66 (12.69)**	**487 (45.77)**	**279 (57.29)**	**144 (29.57)**	**64 (13.14)**
Sexual violence								
Forced sexual intercourse	245 (23.03)	112 (45.71)	73 (29.80)	60 (24.49)	245 (23.03)	108 (44.08)	79 (32.24)	58 (23.67)
Forced to perform other sexual acts	314 (29.51)	156 (49.68)	101 (32.17)	57 (18.15)	301 (28.29)	153 (50.83)	97 (32.23)	51 (16.94)
Sex that degraded/humiliated a woman	258 (24.25)	49 (57.75)	70 (27.13)	39 (15.12)	256 (24.06)	145 (56.64)	69 (26.95)	42 (16.41)
At least one episode of sexual violence	**339 (31.86)**	**199 (58.70)**	**94 (27.73)**	**46 (13.57)**	**331 (31.11)**	**187 (56.50)**	**94 (28.40)**	**50 (15.11)**
At least one of the three forms of GBV	**627 (58.93)**				**594 (55.83)**			

Number and % = number and percentage of participants who reported each item and type of GBV. Column uses 1064 as the denominator and participants could answer yes to multiple items—i.e., the percentage may not add up to 100 for “ever” and in the past 12 months. GBV = gender-based violence; a = these values use the “ever” frequency as a denominator (so that participants only have one response); b = these values use the past 12 months frequency as a denominator (so that participants only have one response). At least one “yes” answer to the items for each type of GBV categorized the person as committing any type of GBV act, but the column percentage may not add up to 100, as the participants answer multiple responses.

**Table 3 ijerph-17-05536-t003:** Differences in the GBV perpetration by educational level and age.

		Types of GBV							
		Emotionally Violent Acts		Physically Violent Acts		Sexually Violent Acts		Any Type of GBV Acts	
Variables *n* = 1064		Lifetime 517 (48.59)	Last 12 Months 488 (45.86)	Lifetime 520 (48.87)	Last 12 Months 487 (45.77)	Lifetime 339 (31.86)	Last 12 Months 331 (31.11)	Lifetime 627 (58.93)	Last 12 Months 594 (55.83)
Age	≤15 (*n* = 196)	113 (21.86)	108 (22.13)	113 (21.73)	109 (22.38)	80 (23.60)	80 (24.17)	132 (21.05)	125 (21.04)
	>15 (*n* = 868)	404 (78.14) ^a^	380 (77.87) ^a^	407 (78.27) ^a^	378 (77.62) ^a^	259 (76.40) ^a^	251 (75.83) ^a^	495 (78.95) ^a^	469 (78.96) ^a^
Education	9–10 (*n* = 815)	407 (78.72)	387 (79.30)	411 (79.04)	385 (79.06)	271 (79.94)	267 (80.66)	490 (78.15)	465 (78.28)
	11–12 (*n* = 249)	110 (21.28)	101 (20.70)	109 (20.96)	102 (20.94)	68 (20.06)	64 (19.34) ^a^	137 (21.85)	129 (21.72)

hypothesis = is there a difference in age and educational status of participants who perpetrated any type of GBV in the “ever” timeframe and in the past 12 months? Pearson’s chi-square (two-tailed); a = *p* < 0.00; Educational level: Grades 9–10 = senior secondary school; Grades 11–12 = preparatory school. GBV = gender-based violence; *n* (%) = number and column percentage in each education or age category for those who perpetrated each type of GBV in the “ever” timeframe and over the past 12 months (column %).

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
