# Peer review of "Gender-Based Violence Perpetration by Male High School Students in Eastern Ethiopia"

_ijerph, 2020, doi:10.3390/ijerph17155536_

Round 1

Reviewer 1 Report

The main problem that I have with the paper is the use of "during a lifetime" as I am not sure can mix adolescents  with young adults.   Perhaps the authors could explain in the methods section why they used this instead of more traditional age groupings. It does appear that there is a large increase of abuse with increasing age.  I would like to see the magnitude of the increase by age, especially if thinking about types of intervention.

Author Response

We have provided a point- by -point response to the reviewer's comments. Please kindly check the attached file.

Reviewer 2 Report

There are still some errors regarding References section style. 

Author Response

There are still some errors regarding the reference section style. 

We have changed all journal-style into italic style with highlighted and track change. Please see page 13, line 406-466.

This manuscript is a resubmission of an earlier submission. The following is a list of the peer review reports and author responses from that submission.

Round 1

Reviewer 1 Report

Add to limitations that As you are interviewing young boys and men who are the perpetrators of GBV you must assume that you will be getting an underestimate of the behaviors that they recognize as wrong.

1st sentence GBV is a global problem for more reasons than having a detrimental effect on health.  For it impedes the choices of girls and women in terms of achievement in school and their life choices.

Is this a reasonable comparison with your study. How Was the UN multi-country study conducted? Were both men and women included in the sample?

On line 47 you refer to ‘these studies’ are you referring to studies based on male or female respondents?

On line 58 are you saying that 70.5% of the men interviewed admitted that were perpetrators of GBV?  This seems quite high, especially compared with the data reported in the 1st paragraph on p.2  Is it due to some aspect of your sample? Line 216:  Why do you think that male data collectors influenced reporting of the data?

The statement starting on Line 60 is not true.  There have been many studies of the prevalence of GBV acts perpetrated by males.  Studies based on reports by males are relatively few because of males are likely not to want to implicate themselves, although this is not what you found.

On line 90 you report that the original questions were about victimization.  T what extent was the wording of the original questions changed? It sounds like the original questionnaire was intended for girls and women.

Did you pilot your questionnaire?

line 109  Can you be more specific? Which 2 periods over the last 12 months?  Was it the same periods of time for all

When comparing with other studies you need to establish the similarities or differences in the samples in terms of gender and age

Reviewer 2 Report

My main concern is the uniqueness and novelty of this study. In this respect, it is unclear whether the scope of this study is substantially different from previous studies that investigated the same topic in Ethiopia (Bekele, 2012; Philpart, 2009). Authors should extensively clarify what does this study add to previous literature.

Much more information should be given about the instruments employed in the study. Authors explain the instruments items, but where does this items come from? If they derive from the cited instruments the origin of each item should be specified.

How was  academic performance assessed?

Information about how questionnaires were administered should be given.

References section doesn´t follow Journal norms.

Reviewer 3 Report

The authors have focused on an important topic and do a thoughtful job of putting their research in perspective and explaining the importance of preventing GBV in Ethiopia. My biggest concern is about the presentation of the results. Some of the results presented were confusing and some of the percentages seem internally inconsistent. I think the results need a careful editing. I also have several other specific concerns and suggestions for the authors to consider.

Specific concerns/suggestions:

Lines 20-24. The denominators are not clear. Is the 37.75% among the full sample or those who perpetrated at least one type? This is an issue in the results section of the text as well. Please be very clear to explain whether the %’s are among the full sample or among those reported any perpetration.

Line 53: typo “this type of societies”

Line 58: typo “three town of Ethiopia”

The 5th and 6th paragraphs in the introduction seem to be making similar points and describe the same two studies on male perpetration in Ethiopia. It seems like they could be combined.

Line 94: when describing the measure of physically violent behavior be sure to indicate who the target was in the question. Was it any female?

Line 119: states “Data quality was checked by providing adequate training and supervision focused on the quality of data during data collection.” Ensuring adequate training and quality during data collection are important but so is checking the quality of the data that were collected. More information about the steps take to ensure the quality of the data collected (e.g., outliers, logical inconsistencies, patterns of missing information, etc.) would be helpful, particularly considering the concerns raised below.

Line 121 was the data collection anonymous? How was confidentiality maintained. What about parental consent?

130 The analysis does not describe how the sampling design and weighting were addressed.

Line 145: Typo: The prevalence of perpetrating acts of any GBV during their lifetime was 58.93% (95% CI: 145 55.97-61.89) and over the last 12 months was it was 55.83% (95% CI: 52.84-58.82; Table 2).

Line 148: The denominator is unclear in the following “During their lifetime, out of those who reported perpetrating multiple types of GBV, 40.44% (95% CI: 148 36.85-44.03) reported perpetrating any GBV acts…” The first part of the sentences seems to state that the denominator is those who perpetrated multiple types but then 100% should have perpetrated any act. I’m not sure what was intended.

Line 160: The denominator is unclear here as well.

Line 188: The phrasing “that degraded/humiliated the women” suggests that it was used to describe the act to the respondent, but this is not included in the measure section. The measure of SV seems inconsistent with the text in the results and table.

The results in Figure 1 do not seem plausible. The categories appear to be mutually exclusive, but the percentages provided total to well above 100%. Also, there is no y-axis in the figure and I’m not sure why the % appears on top of the count in stacked bars. I suggest that the authors consider a different way to present their results.

I have concerns about the accuracy of the results reported in Table 2. For example, there are several examples of behaviors with more respondents saying that they engaged in the behavior “many times” in the past 12 months than in their lifetime. This seems internally inconsistent. Also, some of the “any” composites have counts lower than some of the individual behaviors. For example, if 97 respondents reported many times for “Forced to perform other sexual acts,” how could only 94 report any of the sexual acts many times? If there is an explanation based on how the questions were asked, please explain this in the text or footnotes. 

I found Table 3 difficult to interpret. I think providing column %’s to describe the proportion of those in each education or age category who perpetrated would be more helpful than the row %’s.

Discussion: Results are repeated in the first two paragraphs

Consider citing evidence-based prevention strategies from the WHO like RESPECT https://www.who.int/reproductivehealth/publications/preventing-vaw-framework-policymakers/en/ and INSPIRE https://www.who.int/violence_injury_prevention/violence/inspire-package/en/

Consider citing some of the Violence Against Children Survey data from African countries as background in your introduction or conclusion. https://www.ncbi.nlm.nih.gov/pmc/articles/PMC6476058/.